# Targeting the CD47-SIRPα Innate Immune Checkpoint to Potentiate Antibody Therapy in Cancer by Neutrophils

**DOI:** 10.3390/cancers14143366

**Published:** 2022-07-11

**Authors:** Leonie M. Behrens, Timo K. van den Berg, Marjolein van Egmond

**Affiliations:** 1Department of Molecular Cell Biology and Immunology, Amsterdam UMC, Vrije Universiteit Amsterdam, 1081 HV Amsterdam, The Netherlands; timo.vandenberg@byondis.com (T.K.v.d.B.); m.vanegmond@amsterdamumc.nl (M.v.E.); 2Cancer Center Amsterdam, Cancer Biology and Immunology Program, 1081 HV Amsterdam, The Netherlands; 3Amsterdam Institute for Infection and Immunity, Cancer Immunology Program, 1081 HV Amsterdam, The Netherlands; 4Byondis B.V., 6545 CM Nijmegen, The Netherlands; 5Department of Surgery, Amsterdam UMC, Vrije Universiteit Amsterdam, 1081 HV Amsterdam, The Netherlands

**Keywords:** tumor, antibody therapy, neutrophil, CD47-SIRPα, immune checkpoint

## Abstract

**Simple Summary:**

Immunotherapy aims to engage various immune cells in the elimination of cancer cells. Neutrophils are the most abundant leukocytes in the circulation and have unique mechanisms by which they can kill cancer cells opsonized by antibodies. However, neutrophil effector functions are limited by the inhibitory receptor SIRPα, when it interacts with CD47. The CD47 protein is expressed on all cells in the body and acts as a ‘don’t eat me’ signal to prevent tissue damage. Cancer cells can express high levels of CD47 to circumvent tumor elimination. Thus, blocking the interaction between CD47 and SIRPα may enhance anti-tumor effects by neutrophils in the presence of tumor-targeting monoclonal antibodies. In this review, we discuss CD47-SIRPα as an innate immune checkpoint on neutrophils and explore the preliminary results of clinical trials using CD47-SIRPα blocking agents.

**Abstract:**

In the past 25 years, a considerable number of therapeutic monoclonal antibodies (mAb) against a variety of tumor-associated antigens (TAA) have become available for the targeted treatment of hematologic and solid cancers. Such antibodies opsonize cancer cells and can trigger cytotoxic responses mediated by Fc-receptor expressing immune cells in the tumor microenvironment (TME). Although frequently ignored, neutrophils, which are abundantly present in the circulation and many cancers, have demonstrated to constitute bona fide effector cells for antibody-mediated tumor elimination in vivo. It has now also been established that neutrophils exert a unique mechanism of cytotoxicity towards antibody-opsonized tumor cells, known as trogoptosis, which involves Fc-receptor (FcR)-mediated trogocytosis of cancer cell plasma membrane leading to a lytic/necrotic type of cell death. However, neutrophils prominently express the myeloid inhibitory receptor SIRPα, which upon interaction with the ‘don’t eat me’ signal CD47 on cancer cells, limits cytotoxicity, forming a mechanism of resistance towards anti-cancer antibody therapeutics. In fact, tumor cells often overexpress CD47, thereby even more strongly restricting neutrophil-mediated tumor killing. Blocking the CD47-SIRPα interaction may therefore potentiate neutrophil-mediated antibody-dependent cellular cytotoxicity (ADCC) towards cancer cells, and various inhibitors of the CD47-SIRPα axis are now in clinical studies. Here, we review the role of neutrophils in antibody therapy in cancer and their regulation by the CD47-SIRPα innate immune checkpoint. Moreover, initial results of CD47-SIRPα blockade in clinical trials are discussed.

## 1. Introduction

Cancer is one of the leading causes of death, globally [1]. In 2020, approximately 19.3 million new cancer cases were diagnosed, which is estimated to increase to 28.4 million cases by 2040. Furthermore, almost 10.0 million patients died because of cancer worldwide in 2020 [2]. For many years, surgery, chemotherapy and radiotherapy have been used as the main treatments for cancer. However, durable remissions are not achieved in many cases with these treatments. Therefore, there is a pertinent unmet need to develop new therapies. Immunotherapy focuses on stimulating the patient’s own immune system and recruits immune cells to kill tumor cells [3]. One way to accomplish this is via monoclonal antibodies (mAbs) that target tumor-associated antigens (TAA) [4]. Examples include rituximab directed against CD20 on malignant B cells, trastuzumab against Her-2/neu on, e.g., subsets of breast cancer cells, and cetuximab, recognizing epidermal growth factor receptor (EGFR) that is overexpressed on many epithelial cancers. Antibodies consist of two fragment antigen-binding (Fab) domains and one fragment crystallizable (Fc) region, which can interact with specific antigens and Fc receptors (FcRs) on immune cells, respectively. Anti-TAA mAbs can recruit and stimulate specific immune cells to the tumor microenvironment (TME) [5].

Monoclonal antibodies can have both direct and indirect anti-tumor effects. Direct anti-tumor effects can occur through interference with signaling pathways of growth factors. For example, EGF stimulates tumor cell proliferation, migration and invasion [6]. Antibodies targeting EGFR prevent ligand binding and receptor dimerization, resulting in growth arrest [7,8]. Monoclonal antibodies can also have indirect effects on tumor growth by targeting e.g., the tumor vasculature. During cancer progression, tumor cells stimulate angiogenesis through the production of vascular endothelial growth factor (VEGF). VEGF interacts with VEGF receptor (VEGFR) expressed on endothelial cells, thereby promoting proliferation, migration and survival of vascular endothelial cells [9]. Monoclonal antibodies targeting VEGF or VEGFR inhibit angiogenesis, resulting in suppressed tumor growth in vivo [10,11]. Furthermore, mAbs also act indirectly to opsonize cancer cells and to promote tumor elimination by stimulating the immune system. This may occur in different ways. Firstly, mAbs can stimulate complement-dependent cytotoxicity (CDC), through activation of the complement pathway [12,13]. Complement components subsequently interact to form the membrane attack complex (MAC), which generates pores in the target cell membrane, resulting in tumor cell lysis [12]. Secondly, mAbs can trigger antibody-dependent killing of tumor cells by interacting with FcRs on various immune cells, including NK cells, macrophages and neutrophils. NK cells kill antibody-opsonized cancer cells via antibody-dependent cellular cytotoxicity (ADCC), which involves the exocytosis of cytotoxic granules containing perforin and granzymes and the subsequent induction of cancer cell apoptosis [14]. Macrophages are known to eliminate tumor cells via antibody-dependent cellular phagocytosis (ADCP) [15]. Neutrophils also express various FcRs and can eliminate cancer cells in an antibody-dependent manner by mechanisms described in more detail below [16]. In addition, mAbs can promote T cell-mediated anti-cancer responses by stimulating FcRs on antigen-presenting cells [17].

Another way in which mAbs can promote anti-cancer immunity is by functioning as an agonist for co-stimulatory molecules or as an antagonist for inhibitory receptors. Immune cells express various co-stimulatory receptors, which mediate immune activation essential for tumor eradication. Several agonist antibodies have been developed to stimulate co-stimulatory molecules, such as CD40, GITR and OX40 [18]. These receptors are expressed on T cells, B cells, NK cells, and antigen-presenting cells (APCs), and stimulation of these receptors enhances the anti-tumor immune response [19,20,21]. In addition, therapeutic mAbs can block so called immune checkpoints, which are inhibitory receptors expressed on immune cells that deliver inhibitory signals after interaction with a ligand expressed on either tumor cells or other cells [22]. Under normal conditions, such checkpoints function to maintain homeostasis and prevent autoimmunity. Tumor cells can overexpress these ligands, allowing them to escape immune-mediated killing [23]. For example, overexpression of the immune checkpoint programmed death-ligand 1 (PD-L1) on cancer cells and the presence of programmed death-1 (PD-1) on T cells is correlated with poor disease outcomes in various human cancers [24,25]. Antibodies can be used to block these interactions between the inhibitory receptor and its ligands, which can result in enhanced anti-tumor effects [22]. Currently, most immune checkpoint inhibitors approved for clinical use target the adaptive immune system [26]. Antibodies such as anti-PD-1 (e.g., pembrolizumab and nivolumab), anti-PD-L1 (e.g., avelumab or atezolizumab) and anti-CTLA-4 (e.g., ipilimumab) have been shown to significantly improve patient survival [27,28,29,30]. However, in spite of the clinical progress this has provided, still many patients do not experience durable responses to these antibody therapies [31]. It is likely that the innate immune system is also controlled by immune checkpoints, which could therefore also be attractive targets for therapeutic intervention. Several potential immune checkpoints have been identified on myeloid cells that limit anti-tumor effector functions, and may therefore serve as potential targets for immunotherapy [32]. Currently, the best studied innate immune checkpoint involves signal regulatory protein α (SIRPα), an inhibitory receptor which is more or less selectively expressed on myeloid cells, including neutrophils, macrophages and subsets of dendritic cells. Ligation of SIRPα by its ligand CD47 limits effector functions and thereby tumor cell elimination [33]. This has been extensively described for macrophages. However, more recently a role for SIRPα on neutrophils has been demonstrated. Neutrophils are the most abundant leukocytes in the circulation and play a major role as a first line of defense during infection [34]. In the past decade it has become clear that they also play a significant role in cancer [35,36]. Therefore, neutrophils may also be interesting target effector cells for immunotherapy. In this review, we discuss the role of neutrophils in cancer and describe the contribution of CD47-SIRPα as an innate immune checkpoint for neutrophils. Moreover, the current status of CD47-SIRPα blockade in the clinic will be discussed.

## 2. Neutrophils and Their Role in Cancer

Neutrophils are essential immune cells involved in the initial host defense towards bacterial and fungal pathogens. They are the most abundant leukocytes in human blood, comprising 50–70% of circulating white blood cells [34]. It has been described that in the circulation, neutrophils have a short half-life of only 5.4 days [37]. Therefore, more than 10^11^ neutrophils are estimated to be produced each day in the bone marrow [38]. The role of neutrophils in immunity against pathogens, particularly bacteria and fungi, is well-established. However, only more recently has the important role of neutrophils in cancer has been recognized [35]. The exact functional contributions of neutrophils in cancer appear complex and are not completely understood, as they have been described to stimulate tumor growth, but may also have prominent anti-tumor effects, depending on circumstances [35,39,40,41,42,43,44,45,46,47]. Neutrophils in the TME are also referred to as tumor-associated neutrophils (TANs). In many cases, TANs appear to have largely pro-tumorigenic functions, as a high abundancy of neutrophils in the tumor has been associated with poor prognosis [40,48]. Accordingly, the neutrophil-to-lymphocyte ratio (NLR) in blood has been found to be a prognostic marker for disease outcome [49,50]. Neutrophils may promote tumor progression by functioning as so called granulocytic myeloid-derived suppressor cells (G-MDSCs) [51,52]. G-MDSCs can suppress the immune-mediated destruction of cancer cells by expressing checkpoint molecules or secreting immunosuppressive cytokines [51]. However, the exact mechanisms by which G-MDSC suppress T cell-mediated anti-tumor immunity have not been fully elucidated. Moreover, a number of other mechanism(s) have been proposed by which neutrophils may positively or negatively affect tumor growth and survival. Fridlender and colleagues suggested that the function of TANs is dependent on their activation and differentiation state, and proposed the paradigm of the anti-tumoral ‘N1’ neutrophils versus the pro-tumoral ‘N2’ neutrophils [53]. This distinction is similar—albeit far less well defined—as the one used for tumor-associated macrophages, where classically activated ‘M1’ macrophages are regarded as potent effector cells in tumor killing while alternatively activated ‘M2’ macrophages tune the inflammatory response and promote angiogenesis, tissue remodeling and repair [54]. In addition, it was proposed that neutrophils with an N2 phenotype can be polarized to neutrophils with an N1 phenotype by blocking TGF-β [53]. Identifying ways to stimulate anti-tumor effector functions of neutrophils could therefore be a potential strategy to enhance tumor cell destruction.

## 3. Cytotoxic Effector Functions of Neutrophils

Neutrophils intrinsically possess a variety of cytotoxic effector mechanisms to kill pathogens. To do so, they require stimulation via immunoreceptors, such as Toll-like receptors (TLRs) or C-type lectin receptors (CLRs) that bind unique microbial components known as pathogen-associated molecular patterns. However, cancer cells do not, in general, express such unique molecular patterns for which neutrophils have dedicated receptors. Nonetheless, in the presence of exogenous therapeutic antibodies against TAA, neutrophils use their Fc receptors for the recognition of antibody-opsonized tumor cells. Neutrophils express various FcRs through which they can recognize mAb-opsonized tumor cells (Table 1). Human neutrophils constitutively express FcγRs with a low affinity for IgG, i.e., FcγRIIIb (CD16b) and FcγRIIa (CD32a), as well as the IgA FcR FcαRI (CD89). One study has reported that neutrophils may have a low expression of FcγRIIIa (CD16a) [55]. In addition, a minority of individuals express low levels of FcγRIIb and/or FcγRIIc, based on specific SNPs in the *FCGR2B* or *FCGR2C* gene, respectively, but the functional relevance of these remains to be established [56,57]. In addition, expression of the high-affinity FcγR, i.e., FcγRI (CD64), is increased on G(M)-CSF and/or IFN-γ stimulated neutrophils [58,59]. The most abundant FcγR on neutrophils is FcγRIIIb. However, this is a GPI-linked molecule and does not contain intracellular signaling motifs. Blockade of FcγRIIIb enhanced neutrophil-mediated ADCC of solid cancer cells [60], suggesting that the highly expressed FcγRIIIb acts as a decoy receptor and limits tumor cell killing by neutrophils by competing for antibody binding to other FcγRs. FcγRIIa is a stimulating FcγR, containing an intracellular immunoreceptor tyrosine-based activating motif (ITAM) in its cytoplasmic region. A polymorphic amino acid at position 131 in the FcγRIIa protein modulates the affinity of FcγRIIa and neutrophil killing capabilities towards antibody-opsonized cancer cells for some IgG isotypes, including the currently most frequently used IgG1 and IgG2 therapeutic antibodies [61,62]. Furthermore, selective blockade of FcγRIIa significantly reduced the killing of IgG-opsonized target cells [63], which suggests that FcγRIIa is in fact the principle FcγR receptor involved in ADCC. FcγRI is only expressed on activated neutrophils and has high affinity for IgG. However, its role in regulating ADCC is not clear yet. Specific blockade of FcγRI partially reduced neutrophil-mediated killing of IgG-opsonized Raji cells [64], but in other studies, no effect on ADCC was observed when FcγRI was inhibited on neutrophils [61,64,65]. Neutrophils also express FcαRI, the FcR for IgA. Stimulation of FcαRI on neutrophils with IgA TAA-mAbs results in potent tumor cell killing, which notably occurs in absence of prior neutrophil activation [66,67,68,69,70]. In addition, FcαRI (but not FcγR) signaling results in the secretion of leukotriene B4 (LTB4), which is a potent neutrophil chemoattractant promoting the recruitment of neutrophils to the tumor site [71,72]. Thus, different FcRs may play a role in the recognition of mAb-opsonized tumor cells by neutrophils necessary for tumor cell killing.

After recruitment of neutrophils to the TME and recognition of mAb-opsonized tumor cells, an immunological synapse is created between neutrophils and tumor cells. Neutrophil-mediated cytotoxicity is strictly dependent on the formation of this immunological synapse [73,74]. It has been demonstrated, based on antibody-blocking experiments, that Mac-1 (i.e., complement receptor (CR) 3; α_m_β2, consisting of CD11b and CD18) is an absolute requirement for the establishment of this immunological synapse [74,75]. Cytotoxicity towards antibody-coated tumor cells was absent in Mac-1-deficient neutrophils, compared to normal Mac-1-expressing neutrophils [74]. In addition, immune cells isolated from Mac-1-deficient mice were unable to kill tumor cells ex vivo [75].

Neutrophils have various cytotoxic mechanisms that may be involved in tumor cell eradication. Together with basophils and eosinophils, neutrophils are categorized as granulocytes, named for the abundance of granules in the cytoplasm. These granules are important for their function, as they store mixtures of anti-microbial components and toxic proteases [76]. Neutrophils contain four different subtypes of granules, that can be separated based on their contents: primary, secondary and tertiary granules, as well as secretory vesicles [77]. The primary, secondary and tertiary granules are created at different times during neutrophil differentiation, and therefore some compounds may be present in multiple granule subtypes, while others are mainly found in one type of granule. Primary granules, also referred to as azurophilic granules, contain potent cytolytic enzymes, such as myeloperoxidase (MPO), neutrophil elastase (NE), cathepsins and defensins [78]. Secondary granules, also known as specific granules, and tertiary, or gelatinase, granules have similar contents, both containing matrix metalloproteases. However, secondary granules store lactoferrin, while tertiary granules contain relatively high levels of gelatinase [78]. Secretory vesicles contain human plasma proteins, suggesting that they are created through endocytosis. Secretory vesicles are important because of their receptor-rich membrane, containing for example the β2-integrin Mac-1 [79]. After stimulation, neutrophils release the contents of these granules into the extracellular space. During degranulation, first the secretory vesicles are released, followed by tertiary, secondary and finally primary granules [80]. The different cytotoxic compounds may subsequently contribute to the eradication of tumor cells, as different components isolated from neutrophil granules, including MPO [81], defensins [82], granzyme B [83] and NE [84], were able to kill tumor cells. In addition, different cytokines and chemokines can be released, which can stimulate and recruit other immune cells [85]. Nonetheless, different studies have shown that the disruption of granule release or proteolytic activity did not affect tumor cell killing in vitro [64,65,86], suggesting that neutrophils can kill tumor cells independent of degranulation.

Some studies have suggested that neutrophils may induce tumor cell apoptosis in vitro through the Fas/Fas ligand (FasL) pathway, which is an important cellular pathway involved in the regulation of apoptosis (Figure 1). It has been demonstrated that FasL-expressing myoblasts were able to eliminate rhabdomyosarcoma (Rh) cells in a Fas/FasL-dependent manner [87]. As neutrophils express FasL [88], induction of the Fas/FasL pathway might be one mechanism to induce apoptosis in cancer cells. In vitro inhibition of Fas/FasL signaling resulted in increased tumor growth in a co-culture of neutrophils and A549 lung cancer cells [89]. However, these results have not yet been confirmed by other studies.

Another potential cytotoxic effector mechanism of neutrophils is the formation of neutrophil extracellular traps (NETs) during a process referred to as NETosis. During NETosis, neutrophils can release granular proteins in combination with chromatin, forming an extracellular structure that has been shown to trap and kill bacteria [90]. In general, NETosis is considered a form of active/regulated neutrophil cell death, during which the nuclear envelope and granular membranes disintegrate, followed by rupture of the cell membrane, releasing the NET [91]. Another form of NETosis, termed ‘Vital NETosis’, that does not require neutrophil lysis has additionally been proposed [92,93]. It has been suggested that neutrophils can secrete chromatin via vesicular transport, allowing the neutrophil to survive and perform effector functions even after NET release [92,94,95]. Although NETs have been shown to effectively kill microbes, no studies have shown that NETs can induce tumor cell death. Furthermore, several studies have shown that the presence of NETs in the tumor is associated with worse cancer progression and prognosis [96,97,98,99]. These studies have suggested that NETs promote tumorigenesis by shielding the tumor from immune cells [98], or by enhancing tumor cell motility in response to the interaction between NET DNA and CCDC25 [99]. Further studies are needed to clarify the role of NETs in anti-cancer immunity by neutrophils.

Furthermore, neutrophils are able to efficiently kill pathogens through phagocytosis. During this process, neutrophils undergo morphological changes, allowing them to completely encapsulate microbes, particles or small cells into a vacuole called the phagosome [100,101]. Phagocytosis is an active process; thus, recognition of the target is needed through different receptors, such as pattern recognition receptors (PRRs) and FcRs. After engulfment, the phagosome undergoes a maturation process, allowing it to acquire its cytotoxic properties. The phagosome fuses with neutrophil granules, thereby releasing cytotoxic enzymes into the phagosome, creating a phago-lysosome. This creates an acidic and highly toxic environment, resulting in degradation of the phagocytosed particles [101]. In addition, the activated NADPH oxidase complex produces reactive oxygen species (ROS) in the phago-lysosome. However, neutrophils from patients with chronic granulomatous disease (CGD), which have mutations in components of the NADPH oxidase and thereby do not produce ROS, were equally capable of killing antibody-opsonized cancer cells compared to neutrophils from healthy donors [64,65,86]. In addition, inhibition of the NADPH oxidase by diphenyleneiodonium (DPI) did not affect neutrophil-mediated tumor killing [65]. Thus, at least the NADPH oxidase complex appears not to be involved in tumor killing by neutrophils. It was previously suggested that neutrophils can phagocytose small tumor cells, i.e., B chronic lymphocytic leukemia [102]. However, it was later shown using live-cell imaging and flow cytometry that neutrophils may actually use another mechanism, referred to as trogocytosis, to acquire tumor cell fragments [103], which can lead to the killing of tumor cells [65]. During trogocytosis, neutrophils ingest small pieces of cancer cell plasma membrane in a Mac-1-dependent manner [65,104]. This occurs by an endocytic process in which the tumor material ends up in phago-lysosomes containing granule-derived material such as MPO and lactoferrin. Eventually, trogocytosis can result in a necrotic type of cell death, called trogoptosis [65]. Not only is this mechanism important for the destruction of particularly solid tumor cells [60,61,64,65], but trogoptosis may potentially also cause the release of danger-associated molecular patterns (DAMPs) or tumor antigens, that may further enhance the anti-tumor immune response. It is as of yet not entirely clear how trogoptosis is exactly induced. However, it has been suggested that mechanical forces exerted by the high-avidity state of the Mac-1 integrin play a critical role in the disruption of the tumor cell membrane [65,104]. Interestingly, whereas neutrophils can kill antibody-opsonized tumor cells by trogoptosis, recent evidence demonstrates that tumor cells can resist trogoptosis by membrane repair, a process that is apparently largely mediated via the exocyst complex [105].

In addition to their effector functions in the innate immune system, several studies have suggested that neutrophils may also play a role in activating and regulating adaptive immunity [106,107]. Neutrophils can migrate to lymphoid organs [108], where they can regulate T lymphocyte functions through the release of cytokines, or by acting as antigen-presenting cells [109,110,111]. These different functions show that neutrophils have various ways by which they can stimulate the eradication of tumor cells, obviously making them an interesting and meaningful target effector cell for anti-tumor therapy.

## 4. CD47-SIRPα as an Innate Immune Checkpoint in Neutrophil-Mediated Tumor Killing

In recent years, different mechanisms have been identified that counteract cytotoxic effector mechanisms of neutrophils. For example, inhibitory pathways can limit neutrophil activation and thereby prevent tumor cell killing. Currently, the best studied inhibitory pathway is the interaction between CD47 and SIRPα, which constitutes the focus of this review.

The SIRP family is a multigene family consisting of five members: SIRPα, SIRPβ1, SIRPβ2, SIRPγ and SIRPδ in humans [112]. SIRPα (also known as CD172a, SHPS-1, p84, MFR, MYD-1 or PTPNS1) is an inhibitory receptor expressed on myeloid cells, including macrophages, neutrophils and myeloid dendritic cells, as well as on neuronal cells in the central nervous system [113]. The protein contains three extracellular immunoglobulin (Ig) superfamily (IgSF) domains, consisting of one V-type IgSF (IgV) domain and two C1-type IgSF (IgC) domains, one transmembrane region and an intracellular tail capable of inhibitory signaling (Figure 2) [114]. The intracellular tail contains four tyrosine residues, forming two typical immunoreceptor tyrosine-based inhibitory motifs (ITIM). In addition, the extracellular IgV-domain contains a ligand-binding region, allowing SIRPα to interact with its ligand, CD47 [115].

The CD47 protein (also known as IAP, MER6 or OA3) is a transmembrane glycoprotein expressed on virtually all cells in the body, including both hematopoietic and non-hematopoietic cells [116]. It is a member of the Ig superfamily, and consists of an extracellular IgV-like domain at the N-terminus, a region with five membrane-spanning segments, and a cytoplasmic C-terminus ranging from 3–36 amino acids [117]. CD47 was identified independently on different cell types, resulting in different nomenclature. It was first described as integrin-associated protein (IAP), as it was shown to associate with integrins, e.g., α_v_β_3_, on various cell types [118]. In addition, CD47 was identified as OA3, an antigen overexpressed on ovarian carcinoma cells [119]. As it is now clear that this molecule is expressed on various cell types, and can interact with different proteins, including integrins, thrombospondins (TSP), VEGFR and SIRPs, the current consensus is to refer to it as CD47 [117,120].

The interaction between CD47 and SIRPα was first described in 1999 in mice [121]. Using SIRPα-expressing murine brain cells, CD47 was identified as a binding partner of SIRPα. This was confirmed by anti-CD47 mAbs, which blocked the attachment of various cells to SIRPα-coated substrates [121]. Subsequently, CD47 was also recognized as a ligand for SIRPα in humans [122]. Similarly, in rats a CD47-targeting mAb was identified to prevent the adherence of SIRPα-coated beads [123]. The interaction between CD47 and SIRPα has been analyzed in detail with high-resolution X-ray crystallography and mutagenesis studies [124]. The N-terminal end of SIRPα domain 1 (IgV-domain) consists of four loops, which all contribute to binding of CD47. The N-terminal end of CD47 also forms loops, which are needed for the interaction between CD47 and SIRPα. In addition, CD47 contains a pyroglutamate at the N-terminal, that plays a significant role in the interaction [124,125].

In humans, two allelic variants of SIRPα have been identified: SIRPα_1_ and SIRPα_BIT_ [61]. Within a healthy Caucasian population, SIRPα_BIT_ and SIRPα_1_ homozygotes represent 15.9 and 48.7% of the population, respectively, with 35.4% heterozygous for SIRPα_1_/SIRPα_BIT_ [61]. These variants differ by as much as 13 amino acid residues in the IgV domain responsible for CD47 binding. However, no differences in CD47 binding were observed between SIRPα_1_ and SIRPα_BIT_ [61,126,127]. This may not be surprising, as the polymorphisms occur primarily outside the CD47 binding site [61]. Whereas these polymorphisms could in principle still have an effect on downstream signaling capacities and thereby affect neutrophil effector functions, such as ADCC, no differences were observed in neutrophil-mediated ADCC of trastuzumab-coated SKBR3 cancer cells between neutrophils from donors with the three different genotypes [61]. Thus, it appears that neutrophil ADCC is not affected by the SIRPα genotype.

### 4.1. SIRPα Signaling

To investigate the mechanism of CD47-SIRPα signaling, the immunological synapse between target cells and neutrophils was investigated. It was already established that Mac-1 is essential for the formation of this synapse [74]. However, whether and how SIRPα signaling affects the formation or maintenance of the synapse was not yet clear. During the formation of effector-target interactions, CD47 and SIRPα are both present in the immunological synapse, as SIRPα translocates to the synapse in the presence of CD47, while it is excluded from the synapse in the absence of CD47 [128,129]. Cell–cell contacts between neutrophils and CD47-expressing or CD47-deficient SKBR3 cells were analyzed and indicated that disruption of the CD47-SIRPα axis resulted in the promotion of neutrophil–tumor cell interactions in the presence of tumor-targeting antibodies [65].

After ligation by CD47, ITIM motifs in the cytoplasmic tail of SIRPα are phosphorylated, most likely by Src family kinases. This leads to recruitment of tyrosine phosphatases, in particular Src homology region 2 (SH2)-domain-containing phosphatase-1 (SHP-1) and -2 (SHP-2), which are considered to be principal mediators of SIRPα inhibitory signaling (Figure 2) [130,131]. After recruitment of SHP-1 and SHP-2 to SIRPα, these phosphates undergo conformational changes, allowing them to become activated [132]. The phosphatases can subsequently dephosphorylate various downstream substrates, thereby regulating pivotal intracellular signaling pathways, such as FcR and TLR signaling [65,86,133]. Therefore, different effector functions can be regulated by inhibitory signaling through CD47-SIRPα interactions. In addition, SIRPα might associate with the inhibitory protein kinase Csk and the adaptor protein Grb-2 [134], but the role of these molecules in neutrophil killing has not been explored.

Recently, it was demonstrated that the disruption of CD47-SIRPα interactions resulted in increased Mac-1 integrin activation. The resulting enhancement of cytotoxic synapse formation likely explains the potentiation of cytotoxicity after CD47-SIRPα blockade [104]. Integrin activation can be triggered by various extracellular stimuli, including FcR signaling, which occurs in the context of antibody-mediated cytotoxicity. In the case of β2 integrins (e.g., Mac-1), talin1 and kindlin3 subsequently bind to the intracellular domain of the β chain (CD18), stimulating integrin activation and association with the cytoskeleton [135]. Patients with leukocyte adhesion deficiency type III (LAD3) lack expression of kindlin3 due to a mutation in the gene encoding the kindlin3 protein, *FERMT3*. Using neutrophils of LAD3 patients, it was demonstrated that interference with SIRPα signaling promotes integrin activation in a kindlin3-dependent manner [104]. Thus, it seems that CD47-SIRPα interactions prevent integrin activation in a kindlin3-dependent manner (Figure 2), resulting in less firm cell–cell contacts between neutrophils and antibody-opsonized tumor cells, reduced trogocytosis and, consequently, a virtual absence of trogoptosis in LAD3 neutrophils. Blocking the CD47-SIRPα axis therefore results in enhanced integrin activation, subsequently stimulating trogoptosis of antibody-opsonized tumor cells.

A similar mechanism might affect effector functions of other SIRPα-expressing cells, such as ADCP by macrophages. The interaction of CD47 with SIRPα on macrophages led to suppressed integrin activation and reduced the spreading and engulfment of mAb-opsonized beads [128]. In addition to myeloid cells, SIRPα is also expressed on B1 lymphocytes, a subtype of murine B cells, which produce natural antibodies [136]. Using transgenic mice which lack the intracellular domain of SIRPα and therefore have defective SIRPα signaling, it was observed that B1 cells produce more antibodies when SIRPα signaling is disrupted. In addition, these SIRPα-mutant B1 cells displayed enhanced Mac-1 integrin-dependent migration [136]. Taken together, these studies demonstrate that blocking SIRPα can enhance various Mac-1 integrin-dependent cellular functions, including cytotoxicity and migration, and suggest that the function of the CD47-SIRPα checkpoint may be intimately linked to that of Mac-1.

### 4.2. Neutrophil Effector Functions Influenced by CD47-SIRPα

Neutrophils have various effector functions essential for their role in immunity. As SIRPα signaling can regulate various signaling pathways, different effector functions may be influenced by the CD47-SIRPα interaction. In the 1990s, it was suggested that CD47 may play a role in neutrophil transmigration [137,138,139]. More recently, signaling via the CD47-SIRPα axis has been demonstrated to regulate neutrophil-mediated cytotoxicity [140].

During neutrophil transmigration, neutrophils undergo several steps, i.e., tethering, rolling, adhesion, and transmigration, allowing them to enter tissues from the blood circulation. During tethering, neutrophils form weak interactions with endothelial cells to slow down neutrophils, allowing their rolling across the endothelium. This, in conjunction with other stimuli, induces formation of stronger intercellular interactions resulting in firm neutrophil adhesion to endothelial cells. Neutrophils are subsequently able to cross the endothelial layer and enter the tissues. The first indication that CD47 may play a role in neutrophil transmigration was observed in vitro, since blocking CD47 with anti-CD47 mAbs resulted in delayed fMLP- and IL-8-mediated transmigration of neutrophils across an epithelial monolayer [137,141]. Anti-CD47 mAbs did not disrupt neutrophil adhesion to epithelial cells, indicating that CD47 may affect the transmigration step of neutrophils across the epithelial layer [138]. This effect was mediated by tyrosine kinases, such as Src family kinases and Syk tyrosine kinases, as specific inhibition reverted the effect of anti-CD47 mAbs [141,142]. In vivo studies with CD47-deficient mice suggested a prominent defect in neutrophil extravasation leading to a lethal defect in the clearance of pathogenic bacteria. While this defect in neutrophil migration was clearly linked to β3-integrin function, it is not known whether CD47 was primarily required for pathogen recognition by neutrophils or for the actual migration process itself [139]. CD47 has a variety of well-established binding partners, including integrins, TSP-1, VEGFR and SIRPs. Therefore, it was investigated as to what extend CD47-SIRPα interactions are regulating trans-endothelial/epithelial migration. Anti-SIRP mAbs inhibited neutrophil transmigration across an epithelial monolayer and collagen-coated filters, albeit with different kinetics when compared to anti-CD47 mAbs [143]. Blocking the CD47-SIRPα interaction with a function-blocking peptide, which binds to the CD47 binding domain of SIRPα, resulted in inhibited neutrophil transepithelial migration in vitro [144]. Of note, questions with respect to the specificity of the peptide for CD47-SIRPα interactions can be raised. However, SIRPα-mutant mouse neutrophils, lacking the cytoplasmic region of SIRPα, transmigrated significantly less in vitro when compared to wild type (WT) neutrophils in response to the chemoattractant C5a [145]. In vivo, transmigration of these SIRPα-mutant neutrophils was also slightly delayed when compared to WT neutrophils [145]. This demonstrates that signaling via the intracellular tail of SIRPα may, at least to some extent, controls neutrophil transmigration. Nonetheless, it remains difficult to anticipate how much of an effect SIRPα signaling may have on the overall accumulation of neutrophils in tissues, including tumors, even though such migration may be clearly affected by CD47-targeting agents.

In addition to an effect on neutrophil transmigration, the CD47-SIRPα axis also regulates neutrophil cytotoxicity. Pioneering studies reported by Oldenborg et al. showed that CD47 restricted the clearing of red blood cells (RBC), suggesting that the broadly expressed CD47 functions as a signal of ‘self’ to control the elimination of normal cells by the immune system [146]. In particular, it was found that CD47-deficient RBCs were rapidly cleared, within hours, after their infusion into healthy recipient mice due to phagocytosis by macrophages [146]. For comparison, normal CD47-expressing RBC have a lifespan of 45 days in mice. Thus, it became clear that CD47 essentially functions as a ‘don’t eat me’ signal. It was established in subsequent studies that SIRPα was the inhibitory receptor limiting the phagocytosis of CD47-expressing erythrocytes and that CD47-SIRPα interactions were also restricting phagocytosis and clearance of IgG- or complement- opsonized erythrocytes [131,147]. It should be noted that not only macrophages but also neutrophils are able to eliminate IgG-opsonized RBCs, at least in vitro, and this process is also enhanced after blocking CD47-SIRPα [140]. This principle extends beyond red blood cells, and has now been observed for platelets and other hematopoietic cells, as well as non-hematopoietic cells [148,149,150,151,152,153]. In line with this, the lack of species compatibility between CD47-SIRPα is an important hurdle for xenotransplantation, and, inversely, an exaggerated binding of human CD47 to NOD SIRPα was found to be responsible for the superior engraftment of human tissues in immunodeficient mice in a NOD background [152,154,155]. These findings firmly established the role of the CD47-SIRPα axis in the clearance of normal cells, and also inspired the initial studies to demonstrate its role as an innate immune checkpoint in the antibody-dependent destruction of cancer cells by macrophages and neutrophils [127,156].

### 4.3. The Innate Immune Checkpoint CD47-SIRPα in Cancer

In the clinic, high CD47 expression has been correlated with a worse prognosis of patients with non-small cell lung cancer (NSCLC) [157]. Interactions between CD47 on tumor cells and SIRPα on neutrophils inhibit neutrophil effector functions, allowing the tumor to escape immune surveillance (Figure 3A) [156,158]. Therefore, targeting the innate immune checkpoint CD47-SIRPα could be a potential way to improve current antibody therapies, as it can stimulate neutrophil-mediated tumor killing (Figure 3B). Blocking CD47-SIRPα can be established by various methods: i.e., anti-CD47 mAbs, anti-SIRPα mAbs, or alternative ways, e.g., by downregulating CD47 or by affecting the SIRPα binding site. In vitro, macrophages can eliminate various opsonized solid [159,160,161,162,163,164,165] and hematological cancer [64,156,166,167] cell types via ADCP, which can be further promoted by treatment with anti-CD47 mAbs. Similarly, anti-CD47 mAbs enhance neutrophil-mediated ADCC of solid cancers in vitro, such as neuroblastoma [168]. However, it appears that neutrophils are less capable of killing hematologic cancer cells, and blockade of the CD47-SIRPα axis with anti-CD47 mAbs is not enough to promote tumor elimination. For example, neutrophils were unable to eliminate rituximab-opsonized B cell lymphoma cells [64]. Even when the CD47-SIRPα axis was disrupted using anti-CD47 Fab fragments, neutrophil-mediated ADCC was not improved, although tumor cell elimination was significantly increased when combined with sodium stibogluconate (SSG; an alleged inhibitor of SHP-1) [64]. It is important to note that some anti-CD47 antibodies can by themselves opsonize tumor cells, depending on their ability to still bind Fc-receptors, and hence act as a two-edged sword, i.e., by opsonizing tumor cells for phagocytosis and simultaneously inhibiting CD47-SIRPα interactions. In some cases, it therefore appears that anti-CD47 antibodies were sufficient for killing by myeloid cells, without the need of additional anti-TAA mAbs [161,166,169,170]. Since CD47 is broadly expressed on normal cells, it is highly undesirable to therapeutically use an anti-CD47 antibody with a functional Fc tail, as this would also trigger effector responses against the patient’s healthy cells. As an alternative to anti-CD47 antibodies, anti-SIRPα antibodies have also been studied for their ability to promote tumor elimination. By targeting SIRPα, neutrophil-mediated ADCC of various opsonized cancer cells, such as breast cancer [65], neuroblastoma [168], and colorectal adenocarcinoma [171,172] was promoted.

Blockade of the CD47-SIRPα axis generally only enhances tumor killing in the presence of tumor-targeting antibodies [61,64,127,168]. Consequently, not only enhanced CD47 expression on the tumor, but also decreased expression of TAAs on tumor cells can reduce or preclude neutrophil-mediated killing, as observed, e.g., in neuroblastoma cells of the mesenchymal phenotype that have lost GD2 expression [168].

Tumor-targeting mAbs stimulate neutrophil activation and tumor killing via FcR binding and signaling. As neutrophils express a variety of FcRs, different antibody isotypes can be used to stimulate neutrophils. Currently, most therapeutic mAbs are of the IgG1 isotype, which bind most FcγRs on neutrophils, including the highly expressed FcγRIIIb, which acts as a decoy receptor [60,173]. Treatment with IgG1 mAbs alone can enhance tumor killing by neutrophils. However, combination with CD47-SIRPα blockade significantly enhanced the cytotoxic capabilities of neutrophils [127]. In addition, some IgG2 mAbs are used in the clinic. IgG2 is able to effectively trigger myeloid cells, like neutrophils, at least as effective as IgG1, since it has a high affinity for FcγRIIa, which is the main FcγR involved in ADCC, and lower affinity for the decoy receptor FcγRIIIb [61,173,174]. Targeting various solid tumor cells (A431, A1207, Kyse-30, SAT, Kyse-150, SCC-25) with anti-EGFR IgG2 resulted in increased neutrophil-mediated ADCC compared to IgG1 [174]. Inhibition of CD47-SIRPα by the decreased expression of CD47 on tumor cells further enhanced anti-tumor effects of neutrophils [174]. Besides IgG antibodies, IgA antibodies can also effectively stimulate neutrophils, as they express FcαRI. Several studies have demonstrated that stimulation of neutrophils with IgA mAbs results in an even more potent anti-tumor response than IgG [66,67,69,70]. Furthermore, the killing of IgA-opsonized targets appears independent of neutrophil pre-activation with e.g., GM-CSF or IFN-γ and G-CSF, which is required for IgG-induced ADCC [127,175,176]. Inhibition of CD47-SIRPα further enhanced the destruction of IgA-opsonized tumor cells by neutrophils in vitro as well as in vivo [86]. Moreover, a combination of anti-SIRPα with IgA tumor targeting also promoted neutrophil recruitment to the tumor site in vivo [86], consistent with the known stimulating effects of IgA on neutrophil-recruiting chemoattractants, such as LTB4, which act in an autocrine fashion and may help to form a positive feedback loop [71].

Within bispecific antibodies (BsAb), opsonization and CD47-SIRPα blocking activity can be combined in one antibody, as Fab regions can target different antigens [177]. By combining a TAA-targeting mAb and anti-CD47 or anti-SIRPα mAb, immune cells can be recruited to the tumor and become fully activated by one antibody. For example, the GPC3xCD47 BsAb targets the TAA GPC3, expressed on hepatocellular carcinoma (HCC) cells, and CD47, as well as FcγRs via a functional IgG1 Fc tail [178]. Both in vitro and in vivo, GPC3xCD47 BsAb promoted neutrophil- and macrophage-mediated tumor killing of GPC3-expressing Raji cells. Similarly, the CD47xEGFR-IgG1 BsAb enhanced neutrophil ADCC of EGFR-expressing cancer cells [179]. The CD70/KWAR23 BsAb targets the TAA CD70 and SIRPα [172]. This BsAb significantly enhanced phagocytosis of CD70-expressing cancer cell lines in vitro. Furthermore, CD70/KWAR23 BsAbs limited the growth of Burkitt’s lymphoma cells in vivo [172]. These preclinical studies have demonstrated that blocking the CD47-SIRPα interaction, by either mAbs or BsAbs, may promote tumor cell killing by myeloid cells such as neutrophils and macrophages.

## 5. Targeting CD47-SIRPα to Potentiate Antibody Therapy

The preclinical evidence that inhibition of the CD47-SIRPα checkpoint may promote the efficacy of tumor-directed therapeutic antibodies has prompted the clinical development of a variety of compounds targeting CD47-SIRPα. Currently, different agents, such as antibodies against either CD47 or SIRPα, or other therapeutic biologics directed against CD47, are being investigated for their ability to block the CD47-SIRPα axis to promote tumor reduction. Whereas CD47-SIRPα targeting is often referred to as a method to improve macrophage mediated-phagocytosis, it is clear that neutrophils may also play a critical role as effector cells towards cancer cells during tumor-targeting antibody therapy in general [36,172,180,181,182]. Moreover, neutrophils may also prominently contribute to the enhanced tumor elimination after CD47-SIRPα disruption [172]. In addition, there is accumulating evidence that also adaptive T cell-mediated anti-cancer immunity can be promoted by CD47-SIRPα blockade [183,184]. Clearly, this also sets the stage for a combination of CD47-SIRPα antagonists with PD1–PDL1 inhibitors [185,186,187,188]. Along these lines, there is even initial evidence that CAR-T cell activity may be promoted by CD47-SIRPα inhibitors [189].

### 5.1. CD47-Targeting Agents

Many different CD47-targeting agents have been developed, including anti-CD47 antibodies and SIRPα-Fc fusion proteins. Currently, 24 CD47-targeting mAbs are tested in 72 clinical trials (Table 2). Some trials have already demonstrated promising results, with limited toxicity and good initial indications for anti-tumor efficacy.

Magrolimab (also known as GS-4721 or Hu5F9-G4) is a humanized anti-CD47 blocking antibody with a IgG4 tail modified to prevent Fab arm exchange [190]. As the IgG tail is still functional, at least to some extent with respect to FcγRI binding [62], the anti-CD47 antibody may simultaneously function as an opsonizing antibody. In pre-clinical studies, combined treatment with Magrolimab and trastuzumab resulted in enhanced anti-tumor effects in NSG and C57BL/6 mice that had been xenografted with human SKBR3 cancer cells [191]. Treatment with Magrolimab or trastuzumab alone did not decrease tumor size in vivo. Due to these promising pre-clinical results, Magrolimab was the first in class anti-CD47 mAb that entered clinical trials, and is currently also the most clinically advanced CD47-SIRPα-targeting agent. This was also the first trial in the field with reported results [192]. In the first phase I clinical trial (NCT02953509), 22 NHL patients were treated with a combination of Magrolimab and rituximab [193]. In general, the combination was well tolerated, as adverse events were predominantly of grade I or II, and included anemia (42%) and infusion-related reactions (36%). The occurrence of anemia was expected, as it is an on-target effect of anti-CD47 antibody therapy. The objective response rate (ORR) in NHL patients was 50%, with a complete response (CR) in 36% of patients [193]. This clinical trial was extended into a phase II trial (NCT02953509), in which NHL patients, divided into diffuse large B-cell lymphoma (DLBCL) and indolent lymphoma, were treated with Magrolimab and rituximab. Again, an interim analysis (*n* = 115 patients) showed that treatment was well tolerated, with only 7% of adverse events being grade III or IV. For DLBCL patients, 39% of patients had an ORR, with 20% having a CR. In addition, for patients with indolent lymphoma, an ORR of 66% was observed, with a complete response in 24% of patients [194]. Magrolimab also showed promising results in a phase I trial (NCT02216409) with patients with advanced solid tumors, including colorectal, ovarian, salivary, fallopian tube, and breast cancer [195]. Treatment with various doses of Magrolimab induced mainly grade I and II adverse events, including but not limited to transient anemia (57% of patients), lymphopenia (34%) and hyperbilirubinemia (34%). Moreover, partial remissions were observed in two patients with ovarian/fallopian tube cancers [195]. Treatment with Magrolimab has also been studied in patients with acute myeloid leukemia (AML) or myelodysplastic syndromes (MDS). Monotherapy of Magrolimab in AML patients in a phase I trial (NCT02678338; CAMELLIA) resulted in anemia in 93% of patients, and hemagglutination in 87% [196]. In addition, 73% of patients achieved stable disease, but no objective responses were observed. To enhance Magrolimab efficacy in patients with AML or MDS, treatment was combined with azacitidine, a chemotherapeutic agent used to treat MDS. Combination of Magrolimab and azacitidine in a phase Ib clinical trial (NCT03248479) showed very encouraging responses in both AML and MDS patients, with an ORR of 65 and 91%, respectively [197,198,199,200]. Grade III or IV adverse events in AML patients included anemia (31%), hyperbilirubinemia (19%) neutropenia (19%), and thrombocytopenia (17%) [200]. In MDS patients, grade III or IV adverse events observed included anemia (38%), neutropenia (19%) and thrombocytopenia (18%). These promising results led to the start of multiple phase III clinical trials with MDS and AML patients (NCT04313881, ENHANCE; NCT04778397, ENHANCE-2; NCT05079230, ENHANCE-3). Currently, a large variety of clinical trials are ongoing to investigate the effect of Magrolimab for the treatment of various cancers (Table 2).

Another anti-CD47 antibody is CC-90002, which has a humanized IgG4-PE (S228P and L235E mutation) tail, preventing FcγR interactions [201]. In pre-clinical studies, CC-90002 induced anti-tumor activity in vitro and in vivo against various hematological and solid cancers [202]. In a phase I trial (NCT02641002), patients with relapsed and/or refractory (r/r) AML and MDS were treated with CC-90002. Serious treatment-related adverse events were observed in 82% of patients and included febrile neutropenia (10/23) and bacteremia (4/23). In addition, no objective responses were observed in the treated patients. Due to the lack of a clinically sufficiently encouraging profile, as well as frequent anti-drug antibodies (ADA) development, this program was discontinued. CC-90002 treatment was also investigated as therapy for NHL patients in combination with rituximab (NCT02367196) [203], but this trial also showed low efficacy and was discontinued.

Letaplimab (also referred to as IBI188) is another anti-CD47 IgG4 antibody. Similar to the other anti-CD47 mAbs, Letaplimab was able to promote macrophage ADCP in vitro. In addition, it stimulated anti-tumor effects in NHL and AML/MDS xenograft mouse models in combination with rituximab or azacitidine [204]. In an initial phase Ia clinical trial (NCT03763149), the tolerability and safety of Letaplimab were assessed in patients with advanced or refractory solid tumors or lymphoma [205]. In general, treatment was well tolerated, with mainly grade I or II adverse events. Three out of twenty patients experienced adverse events of grade III or higher, i.e., hyperbilirubinemia, thrombocytopenia or anemia, each in one patient. Currently, five other clinical trials are ongoing with Leraplimab as a monotherapy, or in combination with rituximab, anti-PD-1, or chemotherapy in patients with various cancers, including solid tumors, lymphomas, MDS or AML (Table 2).

Lemzoparlimab (also referred to as TJ011133 or TJC4) is also a fully human anti-CD47 IgG4 antibody. As most anti-CD47 antibodies cause anemia due to phagocytosis of RBCs, Lemzoparlimab was generated to specifically target CD47 on malignant cells while not recognizing CD47 on RBCs, due to unique CD47 binding properties [206]. In a phase I study (NCT03934814), patients with solid tumors were treated with monotherapy of Lemzoparlimab [206]. During this trial, only grade I or II adverse events were observed, including anemia in 30% of patients. In addition, one out of three patients treated with 30 mg/kg Lemzoparlimab had a partial response, and three out of sixteen patients in the trial achieved stable disease [206]. In the same trial, patients with r/r NHL were treated with a combination of Lemzoparlimab and rituximab [207]. Most adverse events were grade I and II, and anemia and thrombocytopenia were observed as one isolated episode. In addition, three out of seven patients had a CR, one had a partial response, and three achieved stable disease. In an ongoing phase I/II clinical trial (NCT04202003), r/r AML and MDS patients were treated with monotherapy of Lemzoparlimab [208]. Most adverse events were grade I or II, but one patient experienced grade III thrombocytopenia. As recruitment is still ongoing, no results are yet available on response rates in this trial.

Another method to target CD47 is with a fusion protein consisting of the N-terminal IgV-domain of SIRPα and a functional Fc region, also known as SIRPα-Fc. These proteins basically function as a decoy receptor and prevent CD47 binding to SIRPα. In addition, the functional Fc tail can interact with FcγRs, to further enhance anti-tumor activity through, e.g., ADCP or ADCC. An example of a SIRPα-Fc in clinical trials is TTI-621, a fully human SIRPα-Fc with a functional IgG1 Fc region [209]. In vitro, TTI-621 was able to strongly bind various tumor cell lines and primary patient tumors [209]. In addition, TTI-621 also bound to cells in peripheral blood, as CD47 is widely expressed on normal cells. In co-cultures, the addition of TTI-621 significantly enhanced macrophage phagocytosis of various hematologic and solid tumors [209,210]. The in vivo treatment of AML xenografted mice with TTI-621 resulted in significantly reduced tumor burden [209]. Similar results were observed with B cell lymphoma xenograft models. Treatment tolerance and adverse events were therefore assessed in a phase I clinical trial (NCT02663518), in which 164 patients with relapsed or refractory hematologic malignancies were treated with TTI-621 alone or in combination with rituximab or nivolumab (anti-PD-1, a checkpoint molecule on T cells) [211]. Treatment was well tolerated by patients until a maximally tolerated dose (MTD) of 0.2 mg/kg. Grade III treatment-related adverse events occurred in 37% of patients and included thrombocytopenia (20%), anemia (9%) and neutropenia (9%). In addition, some indication of therapeutic responses was observed in patients treated with TTI-621 alone or in combination with rituximab or nivolumab. In NHL patients receiving monotherapy TTI-621, the ORR was 10%, with 5% of patients having a CR. For NHL patients receiving TTI-621 and rituximab the ORR was 23%, with a CR in 9% of patients. In HL patients receiving combination treatment with TTI-621 and nivolumab an ORR of 50%, with 25% of patients achieving a CR. TTI-621 monotherapy resulted in an ORR of 20, 13 and 5% in patients with T cell NHL, HL and AML, respectively [211]. Following these results, 35 patients with cutaneous T cell lymphomas (CTCL) or solid tumors received intralesional TTI-621 in another phase I trial (NCT02890368) [212]. Treatment was well tolerated, as no treatment related adverse events of grade III or higher were observed. Rapid responses (median 45 days) were observed and 90% of patients had reduced tumor sizes after treatment with TTI-621 [212]. Thus, these initial phase I trials demonstrate that treatment with TTI-621 does not cause severe toxicities (at the MTD) and has some anti-tumor effects in various cancer types, e.g., CTCL and hematologic cancers.

TTI-622 also is a fully human SIRPα-Fc, consisting of the CD47-binding domain of SIRPα and an IgG4 Fc tail. It was suggested that TTI-622 does not bind to RBCs, unlike many anti-CD47 agents, thereby limiting adverse events such as anemia. In an ongoing phase I trial (NCT03530683), preliminary results were published of 25 patients with r/r lymphoma, who were treated with various doses of TTI-622 monotherapy [213]. In 48% of patients, treatment-related adverse events were reported, mostly being grade I or II. Grade III adverse events observed included neutropenia (9%), thrombocytopenia (5%) and anemia (2%). Objective responses were observed in nine patients, and included two CRs and seven PRs [214]. In this ongoing trial, combinations of TTI-622 with azacitidine or other chemotherapeutic agents are also being investigated in hematologic cancers. Moreover, a clinical trial (NCT05139225) has started in which the toxicity and efficacy of TTI-621 and TTI-622 are being compared in combination with the anti-CD38 antibody daratumumab in relapsing multiple myeloma patients.

ALX148 (also known as Evorpacept) is another SIRPα-Fc fusion protein. More specifically, ALX148 consists of an inactive human IgG1 Fc region that is fused to a modified N-terminal IgV-domain of SIRPα, which enhances CD47 binding [215,216]. As ALX148 has ~50,000× higher binding affinity to CD47 compared to wild-type SIRPα, it prevents SIRPα ligation by acting as a potent decoy receptor. The Fc region is able to interact with neonatal Fc receptors, allowing for extended pharmacokinetics. Contrarily, the Fc tail is unable to bind human FcγRs, preventing targeting of immune cells to normal cells [216]. In pre-clinical studies, ALX148 improved the phagocytosis of OE19, DLD-1, MM1.R, and Daudi tumor cells opsonized with trastuzumab, cetuximab, daratumumab (anti-CD38), and obinutuzumab (anti-CD20), respectively [216]. Mice engrafted with human B cell mantle cell lymphoma were treated with ALX148 or obinutuzumab alone or as combination therapy [216]. Combination treatment significantly inhibited tumor growth when compared to monotherapies. Similar results were observed in mice engrafted with OE19 gastroesophageal tumors, treated with ALX148 and trastuzumab, and mice harboring Raji B cell lymphoma tumors, treated with ALX148 and rituximab [216]. In a phase I clinical trial (NCT03013218; ASPEN-1), 110 patients with advanced or metastatic solid tumors were treated with various doses of ALX148 alone or in combination with pembrolizumab (anti-PD-1) or trastuzumab [217]. All treatments were well tolerated, with four serious adverse events in patients treated with ALX148 alone, five in patients treated with ALX148 and pembrolizumab, and one serious adverse event related to ALX148 plus trastuzumab treatment. The most common serious adverse events were thrombocytopenia and neutropenia. In addition to toxicity, the preliminary therapeutic effects of ALX148 were assessed. Of patients treated with monotherapy with ALX148 18% had stable disease. Combination therapy of patients with head and neck squamous cell carcinoma (HNSCC) receiving ALX148 and pembrolizumab, NSCLC patients treated with ALX148 and pembrolizumab, and patients with gastric or gastroesophageal junction cancer who received ALX148 and trastuzumab, resulted in stable disease in 18, 20, 5, and 21% of patients, respectively [217]. Currently, nine other clinical trials are ongoing in which ALX148 is being given in combination with various mAbs and chemotherapeutic agents to treat patients with hematologic or solid cancers (Table 3). Most of these trials are phase I or II, but also a phase II/III trial has started for the treatment of advanced gastric cancer in combination with ramicirumab (anti-VEGFR) and paclitaxel (NCT05002127; ASPEN-6). Several other CD47-targeting agents are currently also entering the clinical phase (Table 2 and Table 3).

Targeting CD47 on tumor cells allows for simultaneous tumor cell opsonization when the compounds contains a functional Fc tail. However, as indicated above, CD47 is widely expressed on virtually all cells in the body, and, particularly, hematologic adverse events are often observed in patients treated with anti-CD47 mAbs, e.g., anemia, thrombocytopenia, lymphopenia, and neutropenia. In addition, anti-CD47 antibodies may not only disrupt interactions with SIRPα, but also with other CD47 ligands, e.g., thrombosponin-1 or integrins, which could cause other adverse events [192]. Therefore, anti-SIRPα antibodies may in principle provide a better alternative.

### 5.2. Anti-SIRPα mAbs

Since SIRPα expression is much more restricted, with its expression largely confined to myeloid immune cells, it may be easier to saturate. Therefore, lower antibody concentrations may be needed to obtain beneficial clinical responses [192]. Nonetheless, an important aspect to consider is the large homology between SIRP family members. For example, SIRPγ also binds CD47, but is expressed on T cells, and has been suggested to play a role in T cell activation and transmigration in vitro [203]. Thus, the specificity of anti-SIRPα antibodies is key, and if such antibodies cross-react with other SIRP family members, their potential associated effects on safety and efficacy should be considered.

Recently, three anti-SIRPα antibodies have entered the clinical phase in seven clinical trials (Table 4). The first anti-SIRPα mAb entering clinical trials was CC-95251, a fully human IgG1 anti-SIRPα antibody with a K322A mutation, rendering the Fc tail inactive in terms of complement activation, but maintaining FcγR binding capacity. CC-95251, was selected, as it exhibits a high binding affinity to the different variants of SIRPα and blocks binding to CD47 by binding to the CD47-binding domain in SIRPα [218]. In vitro experiments with DLBCL cell lines demonstrated that CC-95251 alone did not significantly enhance macrophage ADCP. However, in combination with rituximab, CC-95251 had a synergistic effect and promoted ADCP of tumor cells. Toxicity was assessed in cynomolgus monkeys, indicating safe intravenous administration and no significant depletion of blood cell counts [218]. Following these results, a phase I clinical trial (NCT03783403) was initiated, in which 230 patients with advanced solid or hematologic malignancies were intended to be treated with CC-95251 monotherapy or in combination with cetuximab or rituximab. Recently, the first interim results of 17 NHL patients treated with CC-95251 and rituximab were published [219]. In these patients, grade III or higher adverse events included neutropenia (53%), infections (24%) and thrombocytopenia (6%). The ORR was 56% and 25% of patients achieved a CR [219]. This trial is still ongoing, and recently another trial (NCT05168202) has been announced, investigating the effect of CC-95251 in combination with azacitidine on r/r AML and MDS.

BI765063 (also referred to as OSE-172) is a humanized IgG4 anti-SIRPα antibody with S229P and L445P mutations, which only binds to one of the major SIRPα polymorphic variants (V1, also known as SIRPα_BIT_) present in the population. BI765063 is reported to be unable to bind SIRPγ, and thus should preserve T cell activation and migration [188]. In vivo, a murine variant of BI765063 promoted ADCC and ADCP of triple-negative breast cancer cells. In addition, anti-tumor effects were enhanced even further in combination with other checkpoint blockades, e.g., anti-PD-L1 antibodies. Analysis of the TME demonstrated that T lymphocytes accumulated in the tumor in mouse models [188]. BI765063 has entered an initial phase I clinical trial (NCT03990233), in which it is used to treat patients with advanced solid tumors as a monotherapy, or in combination with an anti-PD-1 antibody (BI754091). Preliminary results have been presented at ASCO and ESMO meetings. Fifty patients with solid cancer have received monotherapy BI765063 [220]. No dose-limiting toxicities were observed and mostly grade I and II adverse events were reported. Only one patient experienced a grade III infusion-related reaction and none of the patients had anemia or thrombocytopenia as a result of the treatment. One patient showed durable PR, and had increased CD8 T-cell infiltration into the TME upon BI765063 treatment. After two weeks, an increased expression of PD-L1 was measured on the tumor [220]. Thus, combination with anti-PD-1 or anti-PD-L1 antibodies may further enhance clinical benefit. In the same trial, 12 patients were treated with a combination of BI765063 and an anti-PD-1 mAb (BI754091) [221]. Only grade I/II adverse events were reported, and again no anemia or thrombocytopenia were observed. One patient with endometrial carcinoma had a PR and another patient had significant tumor shrinkage [221]. This clinical trial is still ongoing, whereas two other trials (NCT04653142 and NCT05249426) have started, to investigate the toxicity and efficacy of BI765063 in combination with anti-PD-1/PD-L1 and opsonizing antibodies in solid cancer patients.

One other anti-SIRPα antibody has entered clinical trials: i.e., GS-0189, an anti-SIRPα IgG1 with an N197A mutation. In 2020, a phase I clinical trial (NCT04502706) started in which NHL patients were treated with GS-0189 alone or in combination with rituximab. However, no results have been published yet and the development of this agent has apparently been discontinued after nine patients had been treated. Collectively, preliminary results of clinical trials with CC-95251 and BI765063 have suggested the limited toxicity of targeting SIRPα in patients with solid or hematologic cancers. In addition, partial responses were observed in some patients, with the ones in NHL appearing as least as good as with some of the more advanced CD47-targeting agents. Treatment with BI765063 also resulted in enhanced T cell recruitment. Thus, targeting SIRPα may be an interesting alternative to CD47-targeting agents, to block the CD47-SIRPα interaction and to establish meaningful clinical responses.

### 5.3. Alternative Ways to Disrupt CD47-SIRPα Interactions

Besides CD47- or SIRPα-targeting antibodies, the CD47-SIRPα axis can be disrupted in alternative ways, for example by downregulating CD47. Galectin-9 (Gal-9) is a β-galactoside-binding galectin, and has been described for its role in cancer, as loss of Gal-9 is associated with tumor progression and metastasis [222]. However, recently it has been identified that Gal-9 also affects CD47 expression [223]. Associated with this finding, in co-cultures, treatment with Gal-9 significantly enhanced trogocytosis of FaDu cells by neutrophils, but not phagocytosis by macrophages. In addition to downregulation of CD47 on tumor cells, it was shown that the treatment of neutrophils with Gal-9 induced neutrophil activation, such as induced calcium flux, and degranulation, measured by upregulation of CD11b, CD18, CD11c, CD15, CD66b and CD63 on the cell’s surface. In co-cultures with FaDu or Caco2 cancer cell lines, neutrophils were able to kill significantly more tumor cells after Gal-9 treatment [223].

Recently, a small molecule, RRx-001, was identified as a tumor targeting agent, as it also downregulates CD47 on tumor cells [224]. RRx-001 activates the peroxisome proliferator-activated receptor gamma (PPAR-γ), which is a nuclear receptor transcription factor that inhibits Myc by heterodimerizing with retinoid X receptor. Inhibition of the transcription factor Myc subsequently results in downregulation of CD47 [225]. RRx-001 treatment decreased both the expression of CD47 on A549 lung cancer cells, and SIRPα expression on monocytes and macrophages in vitro [226]. Consequently, enhanced phagocytosis of A549 lung cancer, or AU-565, MCF-7, and MDA-MDB-231 breast cancer cells was observed. Treatment of A549-bearing nude mice with RRx-001 resulted in a significant reduction of tumor growth [226]. A phase I trial (NCT01359982) with 25 patients with advanced soluble cancers showed that treatment with RRx-001 was well tolerated with no clinically significant toxicity [227]. In addition, 67% of patients had stable disease and 5% had a partial response. A phase II clinical trial (NCT02489903) showed that RRx-001 is also able to downregulate PD-L1 on small cell lung cancer cells [228]. Moreover, RRx-001 can have a direct anti-tumor effect through epigenetic modulation in multiple myeloma cells [229]. Currently, RRx-001 is tested in various clinical trials [230].

CD47-SIRPα interactions can also be disrupted by modulating enzymatic modifications of the SIRPα-binding domain in CD47. In a FACS-based haploid genetic screen, the gene encoding glutaminyl-peptide cyclotransferase-like (QPCTL, isoQC) was identified to significantly reduce the binding capabilities of SIRPα to CD47 [125,231]. QPCTL is an enzymatic modifier, which adds pyroglutamate modifications to proteins. It has previously been demonstrated that CD47 contains an N-terminal pyroglutamate, which is involved in SIRPα binding [124]. Knockout of QPCTL decreased SIRPα binding in various human cell lines (HAP1, A375, A431, A549, DLD1, and RKO), while the overall expression of CD47 remained unaffected [125]. Similarly, treatment with small molecule inhibitors targeting QPCTL, i.e., SEN177 and PQ912, significantly reduced binding to SIRPα. In a co-culture of human macrophages and anti-CD20 treated Raji cells, the addition of SEN177 significantly increased phagocytosis [125]. Neutrophil-mediated ADCC of cetuximab-treated A431 cells or trastuzumab-treated Ba/F3 cells was significantly enhanced by treatment with SEN177 or knockout of QPCTL as well. These results were confirmed by other studies, showing that SEN177 treatment significantly enhanced ADCP by macrophages and neutrophil-mediated ADCC [232,233,234]. Another QPCTL inhibitor, luteolin, also abrogated the interaction between CD47 and SIRPα [235]. In addition, in co-cultures of H929 or DLD1 cancer cells with mouse bone-marrow-derived macrophages, phagocytosis was significantly improved after treatment with luteolin. To determine the effect in vivo, human FcαRI transgenic BALB/c mice were injected with 1:1 WT and QPCTL knockout Ba/F3 cells [125]. Mice were subsequently treated with anti-Her-2/neu IgA antibodies or PBS. Only in the IgA anti-HER-2/neu treated mice was profound killing of QPCTL knockout cells observed. In addition, an influx of neutrophils into the tumor was observed as a result of anti-HER-2/neu IgA treatment in combination with QPCTL knockout. The specific depletion of neutrophils with anti-Ly6G antibodies abrogated the treatment effect, demonstrating that neutrophils were the main effector cells eliminating QPCTL-deficient tumor cells [125]. These studies demonstrate that alternative ways of targeting the CD47-SIRPα axis may perhaps also have potential to promote tumor elimination. However, more pre-clinical and clinical studies are needed to demonstrate whether these compounds are well tolerated and effective in patients.

Despite the promising preliminary results observed in the various clinical trials targeting the CD47-SIRPα axis, it is important to consider the ways by which tumor cells may adopt resistance against these therapies. Since neutrophils, but also macrophages, require tumor opsonization with anti-TAA antibodies, loss of TAA expression will prevent tumor opsonization and thereby reduce killing by these immune cells. This has already been observed for neutroblastoma cells, where the TAA expression of GD2 can decrease during anti-GD2 mAb therapy [168]. Moreover, tumor cells may upregulate other (perhaps less well defined) checkpoint molecules to limit immune activation and tumor killing. Lastly, tumor cells could escape elimination by preventing immune cell infiltration, by creating an immunosuppressive microenvironment. Thus, although CD47-SIRPα appears to enhance tumor killing, the therapy is dependent on the opsonization of the tumor cells and possibly also the immunosuppressive state of the TME.

## 6. Conclusions

The inhibitory receptor SIRPα, expressed on myeloid cells, limits immune effector functions upon interaction with CD47 on opposing cells. The CD47 protein is expressed on virtually all cells in the body, including hematologic and non-hematologic cells. Notably, tumor cells often overexpress CD47 on their cell membrane to evade immune-mediated tumor cell killing. Disrupting the interaction between CD47 and SIRPα may therefore promote anti-tumor effects by neutrophils and macrophages. Neutrophils are promising effector cells in cancer therapy, due to their high abundance in the circulation and ability to eliminate opsonized tumor cells in vitro and in vivo. After CD47-SIRPα blockade, neutrophil-mediated tumor cell killing by trogoptosis is greatly promoted. This effect is only observed in the presence of opsonizing tumor-targeting antibodies. Currently, many CD47-targeting agents have been developed and a variety have entered the clinical phase. Some compounds have already shown encouraging results, with limited well-tolerated toxicities and promising efficacy. An interesting alternative to CD47-targeting compounds are SIRPα-targeting agents. Due to the broad expression of CD47, side effects such as anemia, neutropenia and thrombocytopenia were frequently observed in clinical trials. In addition, high concentrations were needed to achieve saturation and therapeutic effects. SIRPα expression is much more restricted, and therefore lower antibody concentrations may be needed to observe beneficial effects. Preliminary results with anti-SIRPα antibodies in clinical trials showed that treatments were well tolerated. In addition, some patients had rapid responses, with significantly reduced tumor sizes. Taken together, targeting the innate immune checkpoint CD47-SIRPα has great potential to potentiate antibody therapy in cancer. Exactly how successful CD47-SIRPα targeting drugs can be, and in which patients with what combinations, will undoubtedly soon become clear in this rapidly advancing field.

**Table 2 cancers-14-03366-t002:** CD47-targeting antibodies investigated in clinical trials. Currently, 25 anti-CD47 mAbs are investigated in 73 clinical trials, as mono- or combination therapy for various indications. Table updated until 7 April 2022.

Anti-CD47 mAbs									
Compound	Type	Phase	Trial Identifier	Name	Indication	Treatment	Results		Ref
							Toxicity (≥Grade 3)	ORR (DCR)	
Magrolimab/GS-4721/ (Hu5F9-G4)	anti-CD47 IgG4	1/1b	NCT02678338	CAMELLIA	r/r AML and hrMDS	Mono	72% anemia 6% lymphopenia	AML = 0% (56%)	[196,236]
		1	NCT03248479		AML (65% TP53 mut)	+Azacitidine	31% anemia 19% hyperbilirubinemia 19% neutropenia 17% thrombocytopenia	65% (97%)	[198,199,200]
					int/high risk MDS (13% TP53 mut)	+Azacitidine	38% anemia 19% neutropenia 18% thrombocytopenia	91% (100%)	[198,199,200]
		1	NCT02216409		Solid cancer	Mono	18% lymphopenia 14% anemia 7% Hyperbilirubinemia	2/44 (OvC and FTC)	[195]
		1b/2	NCT02953782		CRC KRAS wt/mut	+Cetuximab	12% anemia 12% hyperbilirubinemia	wtKRAS = 6% (50%) mKRAS = 0% (38%)	[237]
		1b/2	NCT02953509		NHL	+Rituximab	42% anemia 15% neutropenia 36% infusion reactions	NHL = 45% (62%) Indolent = 61% (85%) DLBCL = 36% (48%)	[193,194]
					NHL	+Rituximab + Gemcitabine + Oxaliplatin			[192,193,194]
		1	NCT03558139		Ovarian cancer	+Avelumab			
		1b/2	NCT03869190	MORPEUS-UC	Urothelial & Bladder cancer	+Atezolizumab			
		1b	NCT03922477		AML	+Atezolizumab			
		1	NCT03527147	PRISM	NHL	+Rituximab + Acalabrutinib			
		2	NCT04788043		HL	+Pembrolizumab			
		1	NCT04599634	VENOM	r/r indNHL/CLL	+Obinutuzumab + Venetoclax			
		1	NCT04751383		High-risk & rrNeuroblastoma/rOsteosarcoma	+Dinutuximab			
						+Dinutuximab + surgery			
		1b/2	NCT04541017		CTCL (Myc. Fungoides & Sezary Syndr.)	+Docetaxel + Mogamilizumab			
						Mogamilizumab mono			
		2	NCT04827576		Solid (mNSCLC, mSCLC, mUC)	Mono			
		3	NCT04313881	ENHANCE	Untreated hrMDS	+Azacitidine			
						Azacitidine mono			
		3	NCT05079230	ENHANCE-3	Untreated AML	+Azacitidine + Venetoclax			
						Azacitidine + Venetoclax			
		1b/2	NCT04435691		r/r and nd AML	+Azacitidine + Venetoclax			
		2	NCT04958785		tnmBC	+Paclitaxel- + Nab paclitaxel			
						+Sacituzumab govitecan			
		2	NCT04892446		r/r MM	+Daratumumab			
						+Pomalidomide + Dexamethasone			
						+Bortezomib + Dexamethasone			
		3	NCT04778397	ENHANCE-2	untreated T53mut AML	+Azacitidine			
						phys choice			
		2	NCT04854499		untreated HNSCC	+Pembrolizumab			
						+Pembrolizumab + chemo/Docetaxel			
		2	NCT04778410		newly diagnosed or r/r AML	+Azacitidine + Venetoclax			
						+MEC chemo			
						+CC-486			
		1	NCT05169944		Malignant brain cancer (children & adults)	Mono			
CC90002	anti-CD47 IgG4PE	1	NCT02641002		r/r AML and hrMDS	Mono	36% feb neutropenia 32% thrombocytopenia 29% anemia 18% neutropenia 11% hypokalemia	AML = 0/14 (0/14) MDS = 0/3 (2/3)	[201,238]
		1	NCT02367196		Solid cancer/MM/NHL	+/− Rituximab	38% neutropenia	NHL = 13% (25%)	[203]
Letaplimab/IBI188	anti-CD47 IgG4	1	NCT03763149		Solid tumors and lymphomas	Mono	1/20 anemia 1/20 hyperbilirubinemia 1/20 thrombocytopenia		[205]
		1/1b	NCT03717103		Solid tumors and lymphomas	Mono			
						+Rituximab			
		1b	NCT04861948		Solid tumors (Lung adenocar., Osteosarc., Soft tissue sarc.)	+IBI188 (Anti-PD1)/chemo + Bevacizumab/GM-CSF			
		1	NCT04511975		newly diagnosed hrMDS	+Azacitidine			
		1b	NCT04485052		newly diagnosed or rrAML	+Azacitidine			
		1b	NCT04485065		newly diagnosed hrMDS	+Azacitidine			
IBI322	CD47/PDL1 BsAb IgG4	1a	NCT04338659		Solid tumors	Mono			
		1/1b	NCT04328831		Solid tumors	Mono			
		1/1b	NCT04912466		Hematologic cancer	Mono			
		1a/1b	NCT04795128		Hematologic cancer	Mono			
		1a/1b	NCT05148442		r/r AML and r/r MDS	+Azacitidine			
AO176	anti-CD47 IgG2	1/2	NCT03834948		Solid tumors	Mono			
						+Paclitaxel/Pembrolizumab			
		1/2	NCT04445701		Multiple myeloma	Mono			
						+DEX/Bortezomib			
SRF231	anti-CD47 IgG4	1/1b	NCT03512340		Solid and hematologic cancer	Mono	1/25 feb neutropenia 1/25 hemolysis 1/21 thrombocytopenia 1/21 elevated lipase/amylase	0%	[239]
TG1801/NI-1701	CD47/CD19 BsAb IgG4	1	NCT03804996		B lymphoma	Mono			
						+Ublituximab			
		1/1b	NCT04806035		CD20+ NHL and CLL	Mono			
						+Ublituximab			
Lemzoparlimab/TJ011133/ TJC4	anti-CD47 IgG4	1	NCT03934814		Melanoma	Mono	5% elevated lipase	1/20 (4/20)	[206]
					NHL	+Rituximab	11% neutropenia	NHL = 71% Indolent = 75% DLBCL = 50%	[207]
					Solid and hematologic cancer	Mono			
						+Rituximab			
						+Pembrolizumab			
		1/2a	NCT04202003		AML and ir/hrMDS	+Azacitidine	1/5 thrombocytopenia	1/5	[208]
		1	NCT04912063		untreated AML and hrMDS	+Azacitidine + Venetoclax			
		1	NCT04895410		r/r MM	Mono			
						+Pomalidomide + Dexamethasone			
						+Carfilzomib + Dexamethasone			
						+Daratumumab + Dexamethasone			
		1/2	NCT05148533		advanced, melanoma, GC, HNSCC	+Toripalimab			
ZL1201	anti-CD47 IgG4	1	NCT04257617		Solid cancer and lymphoma	Mono			
HX009	CD47/PD1 BsAb IgG4	1	NCT04097769		Solid cancer	Mono			
		2	NCT04886271		Solid cancer	Mono			
		1/2	NCT05189093		r/r Lymphoma (NHL, Hodgkin, PTCL)	Mono			
AK117	anti-CD47 IgG4	1	NCT04349969		Solid cancer and NHL	Mono			
		1/2	NCT04900350		MDS	+Azacitidine			
		1	NCT04728334		r/r Solid cancer and NHL	Mono			
		1b/2	NCT04980885		AML	+Azacitidine			
		1b/2	NCT05214482		adv. Solid cancer	+AK112 (anti-PD-1/VEGF BsAb) +/− chemo			
		1b/2	NCT05229497		adv. Solid cancer (Ph2: Adv HNSCC)	+ AK112 (anti-PD-1/VEGF BsAb) + chemo			
		1b/2	NCT05235542		adv. solid cancer (Ph2: Adv GEJ or Esoph Cancer)	+ AK104 (anti-PD-1/CTLA4 BsAb) +/− chemo			
IMC-002	anti-CD47 IgG4	1	NCT04306224		Solid and hematological	Mono			
		1	NCT05276310		adv. solid cancer				
SGN-CD47M	anti-CD47 ADC	1	NCT03957096		Solid cancer	Mono			
PF-07257876	CD47/PDL1 BsAb IgG4	1	NCT04881045		NSCLC, SCCHN, ovarian cancer	Mono			
TQB2928	CD47-SIRPα NME	1	NCT04854681		Solid and hematological cancer	Mono			
H4C1/ SHR-1603	anti-CD47	1/2	NCT04588324		Solid cancer	+SHR2150 (TLR7 agonist) + chemo + Anti-PD-1			
STI-6643	anti-CD47 IgG4 (228P)	1	NCT04900519		Solid cancer	Mono			
sB24M	anti-CD47/TNF-α	1	NCT04895566		Severe Pyoderma	Mono (local application)			
IMM0306	CD47/CD20 BsAb IgG1	1	NCT04746131		NHL	Mono			
SHR-1603	anti-CD47 IgG4	1	NCT03722186		Solid cancer and rrLymphoma	Mono			
anti-CD47	anti-CD47	1	NCT05266274		recurr AML after transplantation	+Azacitidine			
Gentulizumab	anti-CD47	1	NCT05221385		Adv solid cancer and NHL	Mono			
		1a	NCT05263271		r/r AML and MDS	Mono			
TQB2928	anti-CD47	1	NCT05192512		Adv solid cancer and lymphoma	Mono			
BAT7104	CD47/PDL1 BsAb	1	NCT05200013		Adv solid cancer	Mono			
SG2501	CD47/CD38 BsAb	1	NCT05293912		r/r hematologic malignancies	Mono			

**Table 3 cancers-14-03366-t003:** SIRPα-Fc fusion proteins investigated in clinical trials. Currently, 9 SIRPα-Fc compounds are investigated in 22 clinical trials, as mono- or combination therapy for various indications. Table updated until 7 April 2022.

SIRPα-Fc									
Compound	Type	Phase	Trial Identifier	Name	Indication	Treatment	Results		Ref
							Toxicity (≥Grade 3)	ORR (DCR)	
TTI-621	SIRPα-IgG1Fc	1/1b	NCT02663518		Lymphoma	Mono	9% neutropenia 20% thrombocytopenia 9% anemia	NHL = 10% (62%) Indolent = 0% (78%) DLBCL = 29% (29–50%)	[211,240]
						+Rituximab	9% neutropenia 20% thrombocytopenia 9% anemia	NHL = 23% (62%) Indolent = 50% (100%) DLBCL = 20% (57%) CTCL = 18% PTCL = 26% Hodgkin = 13% (63%)	[211,241]
						+Nivolumab	9% neutropenia 20% thrombocytopenia 9% anemia	NHL = 50% (50%)	[211]
		1	NCT02890368		CTCL	Mono	None reported	CTCL = 90% (34%)	[212]
					Solid tumors, melanoma, breast ca, soft tissue sarcoma, CTCL	Mono			
						+Anti-PD-1/PD-L1 (nivolumab, pembrolizumab, durvalumab, avelumab, or atezolizumab)			
						+PEG-IFN-α2a			
						+T-VEC			
						+radiation			
		1/2	NCT04996004		Leiomyosarcoma	+Doxorubicin			
TTI-622	SIRPα-IgG4Fc	1/1b	NCT03530683		r/r NHL	Mono	9% neutropenia 5% thrombocytopenia 2% anemia	NHL = 24% (53%) Indolent = 33% (33%) DLBCL = 21% (43%)	
					r/r NHL, r/r MM, newly diagnosed AML T53wt/mut	Mono			
						+Azacitidine +/− Venetoclax			
						+Carfilzomib + Dexamethasone			
		1/2	NCT05261490		OvC	+Pegylated Doxoribicin			
Both TTI-621 and TTI-622		1	NCT05139225		r/r MM	+Daratumumab Hyaloronidase-fihj			
ALX-148/Evorpacept	affinity-enhanced SIRPα-Fc^DEAD^	1	NCT03013218	ASPEN-1	Hematologic and SCLC	+Rituximab	6% neutropenia 3% anemia	NHL = 48% (60%) Indolent = 61% (91%) DLBCL = 38% (48%)	[242,243]
						Mono			
						+Nivolumab			
					Gastric cancer	+Trastuzumab + Ramucirumab + Paclitaxel	44% neutropenia 33% hypertension 22% anemia	72% (89%)	[244,245,246]
						+Trastuzumab	6% thrombocytopenia 6% neutropenia <5% feb neutropenia WBC count	21% (47%)	[217,244,245]
					HNSCC	+Pembrolizumab + 5FU/Platinum	38% neutropenia 31% anemia	39% (89%)	[245,246]
						+Pembrolizumab	Rare (<5%) anemia, thrombocytopenia, AIHA, neutropenia, pancytopenia	20–40%	[217,245,246]
					NSCLC	+Pembrolizumab	Rare (<5%) anemia, thrombocytopenia, AIHA, neutropenia, pancytopenia	5% (40%)	[217]
		1/2	NCT04417517	ASPEN-2	untreated or r/r hrMDS	+Azacitidine	18% neutropenia 18% feb neutropenia 14% thrombocytopenia 9% anemia	55% (80%)	[247]
		2/3	NCT05002127	ASPEN-6	HER-2+ GC/GJC	+Trastuzumab + Ramucirumab + Paclitaxel			
						Trastuzumab + Ramucirumab + Paclitaxel			
						Ramucirumab + Paclitaxel			
		1/2	NCT04755244	ASPEN-5	AML	+Venetoclax + Azacitidine			
		2	NCT04675333	ASPEN-4	HNSCC	+Pembrolizumab + 5FU/Platinum			
		2	NCT04675294	ASPEN-3	HNSCC	+Pembrolizumab			
						Pembrolizumab mono			
		2	NCT05167409		Microsatellite stable rCRC	+Cetuximab + Pembrolizumab			
		1/2	NCT05025800		NHL	+Rituximab + Lenolidamide			
		1/2	NCT05027139		metastatic/inoperable HER-2+ BC/GEC cancer	+Zanidatamab (anti-HER-2 BsAb)			
		2	NCT05167409		MSS CRC	+Cetuximab + Pembrolizumab			
SL-172154	SIRPα-Fc-CD40L	1	NCT04406623		Ovarian, Fallopian tube, peritoneal cancer	Mono			
		1	NCT04502888		HNSCC & CSCC	Mono (intratumorally)			
CPO107/JMT601	SIRPα ECD/anti-CD20 BsAb IgG1	1/2	NCT04853329		NHL	Mono			
IMM01	SIRPα-Fc	1/2	NCT05140811		AML and MDS	+Azacitidine			
IMM2902	SIRPαECD/anti-HER-2	1	NCT05076591		HER-2+ BC and GC	Mono			
DSP107	SIRPα-41BB fusion	1/2	NCT04440735		Solid/NSCLC	Mono			
						+Atezolizumab			

**Table 4 cancers-14-03366-t004:** SIRPα-targeting antibodies investigated in clinical trials. Currently, 3 anti-SIRPα mAbs are investigated in 7 clinical trials, as mono- or combination therapy for various indications. Table updated until 7 April 2022.

Anti-SIRPα mAbs									
Compound	Type	Phase	Trial Identifier	Name	Indication	Treatment	Results		Ref
							Toxicity (≥Grade 3)	ORR (DCR)	
GS-0189	anti-SIRPα IgG1 N297A	1	NCT04502706		NHL	Mono			
						+Rituximab			
CC-95251	anti-SIRPα IgG1 K322A	1	NCT03783403		Solid and hematologic cancer	Mono	53% neutropenia 6% thrombocytopenia	56% (56%)	[219]
						+Cetuximab/Rituximab			
		1	NCT05168202		r/r AML/MDS	Mono			
						+azacitidine			
BI765063/ OSE-172	anti-SIRPα_BIT_ IgG4 S229P-L445P	1	NCT03990233		Solid cancer	Mono	None reported	1/50 PR (HCC)	[220]
						+BI754091 (anti-PD-1)	1 rash maculo-papular	19% (25%)	[220]
		1	NCT04653142		Solid cancer	Mono			
						+BI754091 (anti-PD-1)			
		1	NCT05068102		Melanoma, HNSCC, NSCLC	Mono (Biodistribution, imaging with ^89^Zr-BI765063)			
		1	NCT05249426		HNSCC	+BI754091 (anti-PD-1) +/− Cetuximab/chemo			
					HCC	+BI754091 (anti-PD-1) +/− BI836880 (anti-VEGF/Ang2 BsAb)			

## Figures and Tables

**Figure 1 cancers-14-03366-f001:**
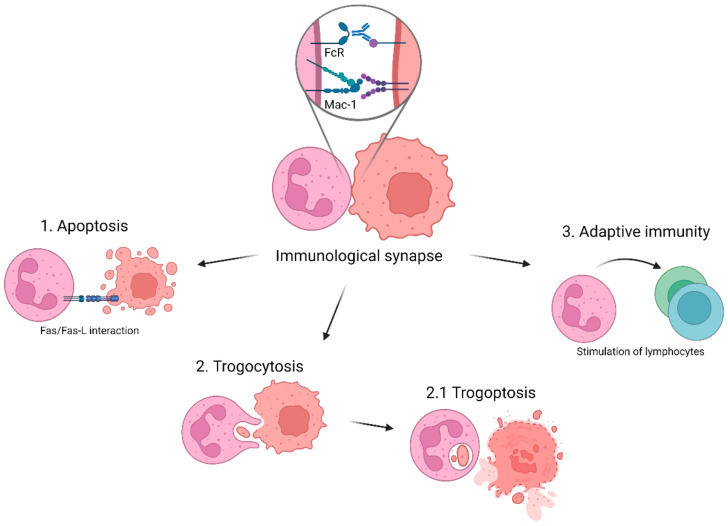
Putative cytotoxic effector functions of neutrophils to kill tumor cells. Neutrophils possess various cytotoxic mechanisms that may play a role in tumor elimination, i.e., induction of apoptosis via Fas/Fas-L interactions, trogoptosis and via stimulation of the adaptive immune system.

**Figure 2 cancers-14-03366-f002:**
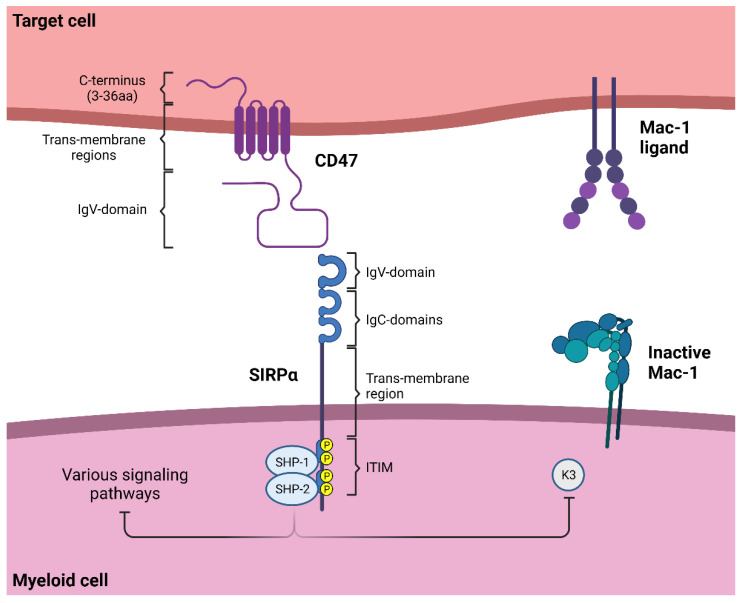
The CD47-SIRPα axis. Interaction between the IgV-domain of CD47 and the IgV-domain of SIRPα results in phosphorylation of the two ITIMs in the intracellular SIRPα tail. As a consequence, the phosphatases SHP-1 and SHP-2 are recruited, which are subsequently activated and able to regulate downstream cellular signaling pathways, e.g., FcR or TLR signaling, by tyrosine dephosphorylation of various mediators. In addition, neutrophil Mac-1 activation is inhibited in a Kindlin3-dependent manner. Abbreviations: K3: Kindlin3.

**Figure 3 cancers-14-03366-f003:**
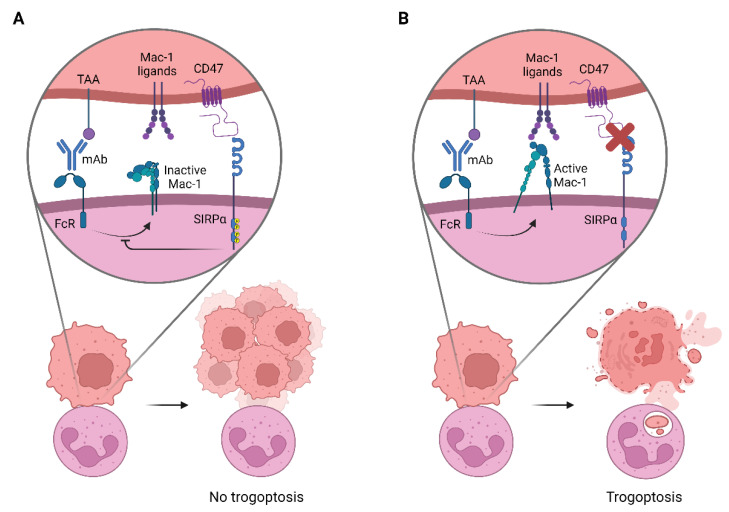
CD47-SIRPα signaling prevents neutrophil-mediated tumor cell killing. (**A**) Ligation of CD47 and SIRPα controls integrin (Mac-1) activation on neutrophils. This subsequently results in less cell–cell contacts between neutrophils and tumor cells, limiting trogoptosis of tumor cells. (**B**) Disruption of the CD47-SIRPα interaction allows Mac-1 activation, resulting in enhanced synapse formation, trogocytosis and eventually trogoptosis of antibody-opsonized cancer cells.

**Table 1 cancers-14-03366-t001:** Fc receptors expressed on neutrophils.

Name	FcγRI(CD64)	FcγRIIa (CD32a)	FcγRIIb (CD32b)	FcγRIIc (CD32c)	FcγRIIIa (CD16a)	FcγRIIIb (CD16b)	FcαRI (CD89)
**Structure**	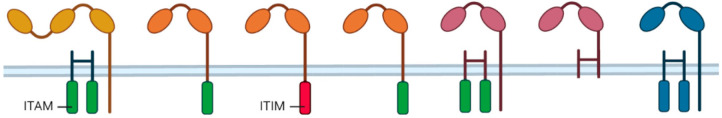
**Affinity**	High	Low	Low	Low	Low	Low	High
**Expression on neutrophils**	Induced ^1^	+	Genotype-dependent ^2^	Genotype-dependent ^3^	−/+ ^4^	+	+
**Class**	Activation	Activation	Inhibition	Activation		Decoy	Activation

^1^ Expression of FcγRI on neutrophils can be induced by stimulation with G(M)-CSF or IFN-γ. ^2^ Expression of FcγRIIb occurs in some individuals (allele frequency ~10.1%) and is dependent on SNPs in the promotor region of the *FCGR2B* gene (promotor haplotype 2B.4) [56,57]. ^3^ Expression of FcγRIIc occurs in some individuals (allele frequency ~11.7%) and is dependent on SNPs in exon 3 and intron 7 of the *FCGR2C* gene [56,57]. ^4^ One study suggested neutrophils may express low levels of FcγRIIIa [55].

## Data Availability

Not applicable.

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
