# Peer review of "Targeting the CD47-SIRPα Innate Immune Checkpoint to Potentiate Antibody Therapy in Cancer by Neutrophils"

_cancers, 2022, doi:10.3390/cancers14143366_

Round 1
Reviewer 1 Report
In this Review the authors describe the role of the interaction between CD47 and the inhibitory signal SIRP-alpha in modulating neutrophil functions in cancer.
The review is interesting and well written. The paper describe the therapeutic opportunities which are under evaluation in clinical trials to exploit this pathway in cancer immunotherapy
Author Response
We thank the reviewer for his/her enthusiasm and for carefully evaluating the manuscript.
Reviewer 2 Report
The review manuscript – Targeting the CD47-SIRPα innate immune checkpoint to potentiate antibody therapy in cancer by neutrophils provide a comprehensive and promising therapeutic review of CD47-SIRPα toward anti-cancer antibody therapeutics. Nowadays, a series of immune checkpoint inhibitors, e.g., anti-PD-1, anti-PD-L1, anti-CTLA-4 have been widely applied to clinical use by targeting the adaptive immune symste. Leonie and colleagues extensively reviewed the role of SIRPα on the interaction with CD47 and described their contribution as an innate immune checkpoint for neutrophils. In general, the review paper is well organized and comprehensively written. A wide rich of journals were cited and enhance the quality of this review. The author firstly showed the importance of neutrophils effects on the tumor microenvironment and proposed several cytotoxic functions to kill tumor cells. Then they showed lots of evidence of differential expression of Fc receptor on recognition of mAb-opsonized tumor cells by neutrophil. The review concentrated on the interaction between CD47-SIRAa, including the mechanism of the signaling in the immunological synapse, neutrophil transmigration regulation, and clinical evidence of innate immune checkpoint. In summary, the author showed clear and load of clinical evidence to show that targeting the innate immune checkpoint CD47-SIRPα can represents a promising therapeutic strategy for cancer treatment.
Even though those results bring an interesting concept that targeting CD47-SIRPα probably can be a potential antiaging therapy, a few major and couple of minor should be taken into consideration.
Major suggestions:
The author showed lots of evidence for the Fc-receptor expressed in the tumor microenvironment and function of neutrophils. However, the relationship between this introduction of Fc-receptor expressed and CD47-SIRPα is weak. So, I suggested to eliminate the cytotoxic effector of Fc-receptor on neutrophils.
It seemed that the inhibiting CD47-SIRPa can enhance the phagocytosis of tumor cells by macrophages. The author focused on the neutrophils, but the specific function and evidence on the neutrophils, especially SIRPa is not robust enough. And these discussions should also be taken more into consideration.
Minor suggestions:
L51: change focusses to focuses.
L100: should be plural outcomes
L337: attachment to attachments
L367: in the promotion of neutrophil ..
L498: miss a comma after normal cells.
Author Response
We thank the reviewer for his/her enthusiasm and for carefully evaluating the manuscript and for pointing out opportunities for improvement.
Point 1: The author showed lots of evidence for the Fc-receptor expressed in the tumor microenvironment and function of neutrophils. However, the relationship between this introduction of Fc-receptor expressed and CD47-SIRPα is weak. So, I suggested to eliminate the cytotoxic effector of Fc-receptor on neutrophils.
Response 1: Neutrophils require Fc-receptor (FcR)-mediated stimulation to be able to recognize and eliminate tumor cells. Thus, in absence of FcR signaling, blockade of the CD47-SIRPα axis does not promote tumor killing. As the FcR stimulation is essential for tumor killing to occur, we included this subsection on FcR on neutrophils in the section about cytotoxic effector functions of neutrophils. We have adjusted this subsection to make this requirement more clear (lines 159-164).
Point 2: It seemed that the inhibiting CD47-SIRPa can enhance the phagocytosis of tumor cells by macrophages. The author focused on the neutrophils, but the specific function and evidence on the neutrophils, especially SIRPa is not robust enough. And these discussions should also be taken more into consideration.
Response 2: Indeed, there is a lot of evidence on CD47-SIRPα blockade on macrophages, which improves phagocytosis of tumor cells. However, more recent data also indicated that this may promote tumor cell killing via neutrophils, which we address in this review. Although in vitro neutrophils are efficiently able to kill tumor cells, in vivo and in the clinic other cells such as macrophages will also play a role in the elimination of tumors. We have added a sentence on macrophages in the introduction (lines 115-117).
Point 3: L51: change focusses to focuses.; L100: should be plural outcomes; L337: attachment to attachments; L367: in the promotion of neutrophil ..; L498: miss a comma after normal cells
Response 3: Regarding the minor suggestions, we have corrected the spelling errors in het manuscript (after adjustments, lines 51, 100, 378, and 509).
Reviewer 3 Report
The manuscript entitled Targeting the CD47-SIRPα innate immune checkpoint to potentiate antibody therapy in cancer by neutrophils by Leonie M. Behrens and coauthors explored the role of the CD47-SIRPα axis in neutrophils functions in cancer. The manuscript is well written and covers a good amount of literature reflecting the importance of exploiting this axis in cancer therapeutics. I suggest the authors also discuss the following questions in the article:
How can cancer cells evade an immune attack and adopt resistance against therapies targeting this axis?
What are the potential limitations for therapeutic targeting of the CD47/SIRPα axis in the clinic?
Can CART T cells be used to target the CD47-SIRPα signaling in cancers?
Author Response
We thank the reviewer for his/her enthusiasm and for carefully evaluating the manuscript and for the relevant questions.
Point 1. How can cancer cells evade an immune attack and adopt resistance against therapies targeting this axis?
Response 1: Despite the shown benefit of blocking the CD47-SIRPα axis, tumor cells may still be able to escape neutrophil-mediated killing. For example, as neutrophils require anti-TAA antibodies for tumor cytotoxicity, downregulation of the TAA may prevent tumor opsonization and thereby killing. In addition, the tumor could perhaps upregulate other (currently less well described) checkpoint molecules to limit immune cell activation and tumor killing. Finally, it has recently been reported that tumor cells can resist neutrophil-mediated ADCC, which involves trogoptosis as we have described, by repair of the plasmamembrane mediated by the exocyst complex (van Rees et al (2022) JITC, DOI: 10.1136/jitc-2022-004820). We have added 2 paragraphs describing these possibilities (lines 879-890 and lines 298-301).
Point 2. What are the potential limitations for therapeutic targeting of the CD47/SIRPα axis in the clinic?
Response 2: In the clinic, targeting the CD47-SIRPα axis may have certain limitations, as already observed in some clinical trials. For example, targeting CD47 with anti-CD47 mAbs requires a relatively higher antibody concentration due to the wide expression of CD47 on all cells in the body. In addition, as healthy cells, in particular erythrocytes, also express CD47, a regular observed side effect of anti-CD47 mAbs is anemia. This limitations may however be overcome by targeting SIRPα instead of CD47, and by inactivating the Fc tail of the antibodies. Lastly, another limitation of targeting the CD47-SIRPα axis in the clinic may be that it needs to be combined with anti-TAA antibodies, as neutrophils, but also macrophages, require these antibodies to eliminate tumor cells. Potentially, combining tumor-targeting and CD47-SIRPα-targeting antibodies can be combined into bispecific antibodies. This topic is addressed in various sections in the manuscript, e.g. in the conclusions at lines 899-909. Additionally, we included an extra paragraph addressing these limitations (lines 879-890).
Point 3: Can CART T cells be used to target the CD47-SIRPα signaling in cancers?
Response 3: Indeed there is initial evidence for this (Chen et al (2022) JITC, DOI: 10.1136/jitc-2021-003737). We have added a phrase in the manuscript to cover this (line 576-581).